# Development of a Training Strategy Aimed at Increasing Veterinarians’ Awareness of the Proper Use of Antibiotics on Rabbit Farms

**DOI:** 10.3390/ani13152411

**Published:** 2023-07-26

**Authors:** Stefania Crovato, Francesca Menegon, Giulia Mascarello, Anna Pinto, Anna Nadin, Gloria Piovan, Guido Ricaldi, Guido Di Martino, Giandomenico Pozza

**Affiliations:** Istituto Zooprofilattico Sperimentale delle Venezie, Viale dell’Università 10, 35020 Legnaro, Italy; fmenegon@izsvenezie.it (F.M.); gmascarello@izsvenezie.it (G.M.); apinto@izsvenezie.it (A.P.); anadin@izsvenezie.it (A.N.); gpiovan@izsvenezie.it (G.P.); gricaldi@izsvenezie.it (G.R.); gdimartino@izsvenezie.it (G.D.M.); gpozza@izsvenezie.it (G.P.)

**Keywords:** antimicrobial use, antimicrobial resistance, antibiotics, rabbit farming, veterinarians training, blended learning course, communication, social research

## Abstract

**Simple Summary:**

Increasing awareness among veterinarians regarding antimicrobial administration to animals is crucial in combatting antimicrobial resistance (AMR). Implementing educational practices serves as a fundamental tool to foster dialogue and discussion among professionals on AMR, facilitating the identification and sharing of strategies for its reduction. This manuscript presents a training project example designed for veterinarians engaged in the rabbit sector. The project utilizes an innovative blended training strategy to encourage communication and discussion among experts, establishing common best practices for the responsible use of drugs on farms. The course’s design was tailored to the specific training needs of veterinarians, identified via a social research activity involving focus group discussions. Participants engaged in interactive activities, such as group classroom exercises and online forum discussions, resulting in the development of an “operational tool”. This tool provides guidance for rabbit veterinarians to implement various strategies in order to reduce antimicrobial usage (AMU) and enhance the adoption of correct breeding management practices. The training activities received positive feedback from the participating veterinarians, highlighting the success of this approach. The structure and efficacy of this training program suggest its potential application in other animal production chains to address AMR effectively.

**Abstract:**

The rabbit sector faces significant challenges with antimicrobial usage (AMU) and antimicrobial resistance (AMR). A focus group involving nine rabbit sector stakeholders identified key issues contributing to high AMU and the need for veterinarians’ training. Participants emphasized the lack of clear legislation, biosecurity standards, and the importance of training on best practices to reduce AMU. To address these concerns, a blended learning course for rabbit veterinarians was organized, focusing on responsible drug use to lower AMU. The course design promoted participant interaction using group exercises and online discussions. The output was an operational tool, encompassing crucial elements to reduce drug dependence, covering housing, environmental conditions, feed, reproduction, disease prevention, diagnosis, and treatments. Validated in veterinarians’ daily practices, the tool proved valuable. The tool, presented as a checklist, assists veterinarians in supporting field activities regarding proper drug use and AMR issues. It also helps farmers address knowledge gaps in breeding management and adopt biosecurity practices for disease prevention. Participants were encouraged to use the tool during farm visits and collaborate with farmers. The project presented in the manuscript is a crucial step towards the development of effective strategies for responsible AMU and AMR mitigation in the rabbit sector and beyond.

## 1. Introduction

Antimicrobial Resistance (AMR) occurs when microorganisms, including bacteria, viruses, fungi, and parasites, become able to adapt and grow in the presence of substances that once inhibited them [1]. As a result, there is an increase in the number of dangerous infections during previously routine treatment, making AMR one of the most urgent threats to public health [2,3].

Among the factors that drive AMR, the selective pressure caused by antimicrobial use (AMU) in humans, companion and food animals, and the environment plays a major role [4]. Therefore, practical measures to minimize the need for antimicrobials (e.g., implement appropriate biosecurity to reduce infections), reduce the use of these drugs (e.g., focus on good practices in therapeutic use and identify alternatives to antimicrobials), and minimize or prevent the spread of AMR (e.g., use good hygiene practices) are part of the global effort to reduce AMR [5]. 

The adoption of the Global Action Plan [6] represents a concerted effort to address the pressing issue of antibiotic resistance (AMR) and its implications for public health. This plan outlines five key objectives, emphasizing the need for enhanced knowledge using surveillance and research, effective communication, education, and training to improve awareness and understanding of AMR and to optimize antimicrobial use in both human and animal medicine. According to the Food and Agriculture Organization (FAO), education and training play pivotal roles in translating knowledge into practical implementation, warranting the inclusion of AMU and AMR as integral components in professional education, postgraduate training, and continuing education within the food and agricultural sectors [7]. In particular, recent studies have highlighted the necessity to bolster training and communication interventions among rabbit farmers and veterinarians [8,9,10,11].

Within the Italian context, the rabbit industry assumes substantial significance, annually producing approximately 24.5 million rabbits for meat production, ranking third in the European Union after Spain and France [12]. Notably, 46.5% of these rabbits originate from the competence area of the Istituto Zooprofilattico Sperimentale delle Venezie (IZSVe) in northeastern Italy [13]. However, the sector encounters critical challenges concerning antimicrobial consumption [14]. Rabbits reared for meat in industrial farms exhibit the highest rates of antimicrobial usage compared to other food-producing animals [15,16,17], leading to alarming rates of AMR within the industry [18,19].

The primary objective of this study is to comprehend and update the training needs of veterinarians operating in the rabbit sector. Additionally, the study aims to devise an innovative training strategy to foster communication and discussion among experts, thereby facilitating the establishment of common best practices to promote the responsible use of drugs on farms. By addressing these concerns, the study aims to contribute to the broader global efforts in mitigating AMR and ensuring the sustainability and welfare of the Italian rabbit industry.

## 2. Materials and Methods

The study used social and educational research methodologies as follows:Target analysis: in-depth analysis of rabbit sector veterinarians’ training needs by means of the focus group technique.Training activity: realization of the training activity using the blended learning approach to enhance the participants’ experiences. Creation of an operational tool for veterinarians’ work activities on farms.Training activity evaluation: assessment of the participants’ learning and satisfaction with the training via a semi-structured questionnaire.

### 2.1. Target Analysis

Qualitative data were collected using the focus group technique [20]. The focus group involved nine stakeholders: three veterinarians working in the rabbit production industry, two freelance veterinarians, two employees of the animal feed industry, one agronomist in the private sector, and one veterinarian of the National Health System (NHS). Joining the meeting was on a voluntary basis. The focus group discussion explored the following topics: (a) critical issues in the use of antibiotics in rabbit breeding, (b) factors that can favor the reduction in drugs in farms, and (c) the training needs of veterinarians. The focus group was held in September 2017; the discussion lasted about one hour and was audio-recorded and manually transcribed. Data collected were analyzed with a manually coded thematic analysis based on the interview guide [21]. To comply with the General Data Protection Regulation (EU) 2016/679, a paper document with a privacy agreement was signed by participants before the focus groups began.

### 2.2. Training Activity

A blended training course entitled “Enhancement of rabbit production through the development of shared health training” was implemented for veterinarians working in the rabbit sector. The objective of the training course was to enable participants to acquire knowledge and technical skills on the main issues regarding the correct use of drugs in breeding and fattening rabbit farms. The training activity was developed using a social-constructivist theoretical model, which prompts the cognitive impact of interpersonal interaction and participants’ self-reflection on their professional experiences [22]. From a methodological perspective, a blended learning model was adopted, combining traditional face-to-face lectures, cooperative learning, and an online learning environment. The training course was designed by a multidisciplinary team of experts in education and veterinary medicine from university and public health institutions. 

The classroom lectures were developed using expert reports on the topics of the course, round tables discussion with experts, presentation of case studies, work in small groups on clinical and management issues in rabbit farms, and clinical cases and discussions with experts. In addition, online learning environments (Moodle web application) were used to support the activities in order to enhance interaction “beyond the classroom” and foster the achievement of learning objectives [23]. The key lies in the increasingly active role that the learner takes on in educational pathways: from the centrality of the contents and experts to the centrality of the participants, their experience, and their self-regulated learning processes. We adopted a perspective that recognizes competence linked to learning by a dynamic relationship: learning resulted from the activity and not from memorization of contents. The 22 h blended course was carried out between May and June 2019. The topics were divided into five modules presented in Table 1. 

In addition to lectures with experts, each module included a “classroom exercise” dedicated to elaborating the concepts covered in the classroom (from theoretical or regulatory concepts to their practical application in the field). This led, at the end of the course, to the development of an operational tool to address the main rabbit diseases (enteric, respiratory, cutaneous). The aim of the tool, developed collaboratively by learners and experts via best practice and operational strategies sharing, is to make available to veterinarians a kind of checklist that is useful for dealing with main rabbit diseases.

Twenty veterinarians attended the course, and they were selected following two criteria: (1) their primary professional field was the rabbit-production supply chain; (2) they worked in northern Italy. Moreover, the participants were selected to maintain a fair representation of both public and private veterinarians, as both groups operate in the rabbit supply chain. 

### 2.3. Learning and Satisfaction Assessment 

Each course activity (discussions, classroom exercise, individual exercise) included a time for formative assessment based on experts’ feedback and peer-to-peer interaction. Specifically, the final individual exercise tested the operational tool developed during the course. Participants described a clinical case faced in daily practice and the interventions identified for its solution (structural/managerial and/or therapeutic). Then, suggestions of possible adjustments and additions to the operational tool were collected in the perspective of continuous improvement, typical of the experiential learning cycle [24].

An online course satisfaction questionnaire was administered at the end of the course. The questionnaire included questions on a 1–5 Likert scale (1 = not relevant; 5 = very relevant) concerning the relevance of the topics covered, the educational quality of the program, the usefulness of the course for veterinarians’ continuing professional development, and the compliance with scheduled deadlines. Open-ended questions were asked to collect proposals for improvements in the planning and implementation of the course and to determine if participants wanted to undertake initiatives aimed at reducing AMR in rabbit farming.

Using Windows Excel software (version 2013), measures of central tendency (mean) and dispersion (standard deviation) were performed to analyze quantitative data.

## 3. Results

### 3.1. Focus Group Results

The information collected during the discussions was summarized and reported in Table 2.

### 3.2. The Training Course and the Development of the Operational Tool

The activities were designed to facilitate interaction among participants using group classroom exercises and online forum discussions. In particular, classroom exercises scheduled for each thematic module promoted dialogue and discussion among veterinarians on possible strategies for reducing AMR. At the end of the course, an operational tool was realized and validated by all participants. The tool is a checklist that can be used in the field and is a synthesis of all the discussions and professional reflections communicated throughout the course. It contains all the practical and theoretical elements that emerged in the classroom exercises and that are considered crucial in the veterinarians’ daily practices. The tool was composed of two parts: the first part contained Farm Characteristics and its Management, including the sections Farm Environment and Prophylaxis Measures, Feeding Management, and Reproductive Stage; the second part included Disease on the Farm, including Section Diagnosis and the Selection of the Most Appropriate Intervention. Each section contains a list of selected variables in the form of a checklist that can be used on the farm and has space where useful information and notes can be recorded and annotated. The tool is the collective product of an experiential learning model based on four cyclical phases: experiencing, reflecting, thinking, and acting. It can be employed by veterinarians to support their activities related to the correct use of drugs on farm and the issues related to AMR. It can also be useful for farmers to manage knowledge gaps in breeding management and to promote the correct adoption of biosecurity practices to prevent diseases. Course participants were advised to bring the tool with them during their visits to farms and use it together with the farmers if needed. The operational tool is available in Appendix A.

### 3.3. Participants’ Evaluation and Feedback on the Training Course

The training activity was assessed by the realization of a practical and individual exercise and via a satisfaction questionnaire. The individual exercise required the application of the operational tool to a practical case that the participants faced during their work. They were asked to describe any possible interventions (structural/managerial and/or therapeutic) and to provide the following information: characteristics of the farm and its management, problems identified, and observations and proposals for intervention. The development of the exercise envisaged the use of the operational tool (Appendix A). Participants each worked on their case study and presented their work, which the course’s scientific director evaluated. All participants worked profitably on the test, passing it with positive results. Table 3 reports some feedback on the application of the operational tool collected after the exercise.

The evaluation of participants’ satisfaction showed that the training activities were greatly appreciated, highlighting that the topics covered were relevant and useful for the participants’ knowledge needs (m = 4.68; st. dev 0.48) (Table 4).

The open answers collected are summarized in Table 5.

## 4. Discussion

A widespread lack of knowledge of AMR and prudent AMU in the farming sector has been well-documented [14,25]. Farmers often perceive numerous benefits but relatively few risks associated with AMU [26]. Several studies in the literature have investigated farmers’ perceptions of AMU and AMR [26,27]. While some farmers consider their AMU as adequate, they believe a reduction of 20–30% is possible [15]. In contrast, veterinarians have increasingly adopted alternative strategies to reduce AMU [28] and promote more responsible antimicrobial practices. Veterinarians serve as influential sources of information for farmers [29,30,31]; most likely, their collaboration has been associated with a reduction in AMU [29,32], including the use of critically important antimicrobials in dairy farms [32].

In investigating the rabbit sector’s challenges, a focus group approach was employed, facilitating immediate critical commentary and comprehensive discussions to address complex issues [33]. Participants highlighted the sector’s limited economic resources and market uncertainty, making farm renewal investments challenging. Additionally, the need for standardization in biosecurity, health management, and drug usage further complicates farm management (Table 2).

The focus group emphasized the importance of developing a public-private network and promoting collaboration among rabbit sector stakeholders as key strategies to reduce AMU (Table 2). Transparent communication and information sharing between the public and private sectors can enhance awareness and stimulate solutions to complex problems, such as vaccine development, facility modernization funding requests, research advancements, and biosecurity standardization. Integrating the expertise of public veterinarians (legislation, epidemiology, best farming practices) with private veterinarians’ competencies (medical skills, pharmacovigilance, practical awareness) is vital, particularly in sectors including the rabbit industry, affected by declining meat consumption and economic crises [12,34]. Moreover, if these two worlds convey the same message to the farmers, it is easier for the message to digest.

Interestingly, the strategies identified by the participants to address technical issues were all cross-sectional. They included radical management changes, such as implementing integrated supply chain models, improved economic forecasting, and developing communication strategies to influence consumer purchase habits (Table 2). For instance, in the egg production sector, consumers in developed countries prefer cage-free eggs [31,35]. These consumers are willing to pay more for eggs produced in cage-free systems that are perceived to comply with higher animal welfare standards [36]. These trends prompt farmers to invest in farm facilities and housing systems and improve farm management.

The blended learning approach adopted in the study facilitated highly interactive participation, effectively achieving the course’s learning objectives. Active participation played a central role in the success of the training and awareness-raising activities for the professionals involved in the administration of drugs on rabbit farms, particularly farmers and NHS professionals. In fact, classroom discussion sections frequently over-ran their allocated time, sometimes exceeding the time limit by one hour and a half, making the participants evaluate the “time” factor lower than the other factors in their course feedback (Table 4).

The approach of community-based participatory research emphasizes empowerment, shared decision-making, and social transformation. The strength of this approach is discovering knowledge in a collaborative, cooperative manner amongst people with a range of different experiences. Veterinarians involved in the blended course were the co-creators of the knowledge, and the result was the creation of the operational tool oriented to guide rabbit veterinarians using the different strategies to reduce AMU, enhancing the application of correct practices for breeding management. Thanks to the positive impact of the training activities developed and experimented with the rabbit veterinarians, this successful training approach and structure could be repeated in other animal production chains.

### Limitations and Future Developments of the Study

Although the number was consistent with the chosen methodological approach, the small number of participants involved is a possible limitation of the study. Since the goal was to foster a high level of interaction among participants, a larger number could not be considered. A possible solution could be to replicate the experience with a larger number of working groups or with separate training sessions (course editions).

Another limitation was related to the time reserved for evaluating the learning impact, which we used to measure the effectiveness of learning. More time availability would have allowed an investigation after one or even two years, fostering a deeper perspective on the usefulness of the operational tool created.

The operational tool developed refers solely to the rabbit sector. Therefore, new courses and training activities should be implemented to create tools targeted to the other breeding sectors. Data on veterinarians’ perceptions, training needs, and feedback on the operational tool should be updated to confirm the operational tool’s usefulness.

## 5. Conclusions

The study highlights the need to enhance training activities and to establish a public-private veterinarians network, alongside fostering collaboration between farmers and veterinarians, as essential measures to raise farmers’ awareness of responsible antimicrobial usage (AMU). Employing a focus group methodology proved to be a valuable and effective approach for in-depth discussions and investigation of risk factors related to antibiotic usage within the rabbit sector. Furthermore, the training activities conducted with rabbit veterinarians received highly positive feedback, leading to the development of an operational tool applicable to veterinarians’ on-farm practices. Consequently, the training approach and course structure hold promise for replication in other animal production chains. By applying similar training interventions and fostering effective stakeholder collaboration, the collective efforts to combat antimicrobial resistance (AMR) and promote responsible AMU can be effectively extended to various sectors of animal production.

## Figures and Tables

**Table 1 animals-13-02411-t001:** Modules and specific topics covered in the training course.

Modules	Topics Covered
AMR, animal health, and environmental sustainability	- Introduction to AMR, animal health, and environmental sustainability - Presentation and discussion on the results of social surveys on Italian veterinarians’ opinions, perceptions, and the information needed on AMR - Use of antibiotics in rabbit farms and AMR: discussion among professionals on experiences and critical aspects
2. Appropriate diagnostic tools and interventions on farm	- Use of diagnostic laboratory tests to support rational therapeutic choices in rabbit farming - Interpretation of biochemical parameters from blood samples - Pharmacokinetics in the treatment of the main rabbit pathologies (enteric, respiratory, cutaneous) - Antibiotic therapy protocols used in the most common rabbit pathologies - Classroom exercise: introduction to the development of the operational tool to address the main rabbit diseases (enteric, respiratory, cutaneous)
3. Animal welfare and biosecurity measures	- Animal welfare and structural husbandry interventions in rabbit farms - Biosecurity measures in rabbit farms - Correlations of AMR with risk factors for AMU and for biofilm development - Classroom exercise: design of the operational tool (checklist) to address the main rabbit diseases (enteric, respiratory, cutaneous)
4. Regulatory issue	- Use of antibiotics: overview of the regulatory framework - Criticalities in the use of drugs in rabbit farming: the point of view of the veterinarians of the public sector (NHS) - Criticalities in the use of drugs in rabbit farming: the point of view of the veterinarians of the private sector - The antibiotic in rabbit farming: between theory and practice - Classroom exercise: realization of the operational tool (checklist) to address the main rabbit diseases (enteric, respiratory, cutaneous)
5. Communication strategy to promote animal welfare and AMR management	- Definition of a shared communication strategy to inform farmers on best practices for AMR reduction - Individual exercise: assessment of the operational tool (checklist) usefulness and validity through application to a practical case study

**Table 2 animals-13-02411-t002:** Information collected during the focus group, divided by topic of discussion (*n* = 9).

Topics	Information
Critical factors limiting the correct use and the reduction in antibiotic use on rabbit farms	- Unclear legislation on drug and galenical (e.g., withdrawal periods, correct use of specific molecules). - Low economic possibility for farmers to restructure facilities. - Meat market uncertainty and high production costs do not allow planning investments. - Absence in the rabbit sector of specific biosecurity standards and measures to protect rabbit health (e.g., vaccinations, alternative breeding systems). - Lack of consumer pressure on the market to have rabbit meat products with specific standards and measures to reduce antibiotic usage in breeding - Lack of training activities for veterinarians and farmers targeted to different farming methods and practices to foster drug reductions
2. Factors that can accelerate and promote the progressive reduction in drug usage in rabbit breeding	- Strengthen farmers’ networks and promote the development of a different organizational model (similar to the French model). - Implement better market forecasts for purchase, consumption, and meat price trends. - Promote more efficient housing and management solutions (e.g., reproductive cycle changes and evaluation of production costs). - Standardize the interpretation of the legislation at the national level. - Foster a closer collaboration between the rabbit sector’s stakeholders: breeders, NHS veterinarians, and slaughterhouse employees. - Develop a communication strategy to sensitize consumers on breeding methods and promote the conscious purchase and consumption of rabbit meat.
3. Useful topics in veterinarians’ training course	- Correct drug usage: treatment indications, pharmacokinetics, interactions with water quality. - Regulations: updates and developments. - Introduction of new technologies on farms. - Insights into the use of autogenous vaccines and vaccine treatments in general.

**Table 3 animals-13-02411-t003:** Participants’ feedback on the application of the operational tool to a practical case study.

Question	Feedback
How do you think the operational tool will be useful in your daily work? Briefly explain your reasons	- Useful as a guide for administering antibiotics properly - Useful to reflect on the issue and detect critical factors - Useful for sharing opinions and practices among stakeholders in the rabbit sector

**Table 4 animals-13-02411-t004:** Participants’ assessment of the training course (*n* = 19; question “How do you evaluate the following aspects of the course you attended? Likert Scale 1.5: 1 = not relevant; 5 = very relevant).

Aspects Evaluated	Mean	St. Dev.
Topic relevance	4.68	0.48
Educational quality	4.58	0.51
Useful for your training/updating	4.58	0.69
Course timing	3.42	0.69

**Table 5 animals-13-02411-t005:** Feedback was collected using the open answers to the evaluation questionnaire.

Question	Feedback
What new insights into professional reflection did the discussion with your colleagues provide you with?	- Importance of establishing a lasting dialogue and collaboration with colleagues, both for a mutual understanding of the problems faced by each category and to better organize common actions aimed at reducing drug usage - Importance of water quality as a parameter determining therapeutic efficacy
What would you have liked to further explore during the course?	- Legislative insights, methods of control in breeding, antibiotic residues in the water, management methods, and improvement in breeding; - More time was spent on the discussion of practical cases during the classroom exercises.
To address antibiotic resistance phenomena, what actions do you propose, considering the information provided by the course?	- Improvement in farm management, with a reduction in the use of medicated feed (for prophylaxis and metaphylaxis), - Development of training/information interventions for all the professional figures involved.
Other suggestions, comments, and proposals useful for improving the IZSVe Training Service	- Framework on the farmer’s production costs - Continuity of professional updates

## Data Availability

The data that support the findings of this study are available from the corresponding author upon reasonable request.

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
