# Peer review of "Development of a Training Strategy Aimed at Increasing Veterinarians’ Awareness of the Proper Use of Antibiotics on Rabbit Farms"

_animals, 2023, doi:10.3390/ani13152411_

Round 1

Reviewer 1 Report

Dear authors,

thank you for submitting the manuscript. You address some important questions concerning AMU and AMR in rabbits, and also concerning methods to communicate. Unfortunately, in my eyes, you included too many different topics in this one paper and do not go deep enough to fulfill the readers' needs. I therefore, advise you to separate the parts survey, focus groups and the recommendations for better education in separate manuscripts and add more detailed methods and results to each of the topic.

Here, some aspects I noted during my reading:

Abstract: Please add the explanations for the terms AMU and AMR

Point 2.1: Please provide a bit more information on the questions in the questionnaire: How many, which topics. Why do you call it semi-structured if only closed-ended questions and Likert scales were used?

Point 2.2: How did you transcribe the audio records?

2.3: Was the training tool developed only for this study? If yes, please provide a bit more information: Who was involved in the development? How long did it take? ppt-slides or videos or...? If this tool was developed in another context, please describe this also shortly

I am completely missing the description of the data analyses in points 2.2 to 2.4 Please add.

Figure 3: Consider using Diverging Bar Charts

3.2: How did you define what a leading information was?

Reviewer 2 Report

Dear authors,

I reviewed your manuscript with much interest and believe it provides valuable insights into a sector (rabbits) which is according to my experience, little investigated. However, I still have some remaining questions and comments with the current version of the manuscript.

Simple summary:
Please check this part again. For me it was very difficult to read and understand. Especially the part where veterinarians are the co-creaters of the knowledge is clear now but it was not when I read the manuscript the first time.

Introduction:
I believe in general the intro focusses to much on the issue of AMR in humans. And although very important, this is not really relevant for the paper.
L51-57 can be ommited I believe
L58-59: I do not 100% agree. AMR is a natural proces. And yes AMU is fastening the process but it is not the origin.

L60 implies that just using new AM (if there would be any) is a solution. Do not think the authors want to state this?

L79-85: I would prefer more details. The location of the authors is for me not a good reason to conduct research there. We know nothing as a reader on the size of rabbit production in Italy. Maybe like with other animal production, the majority is located in the North anyway so this is not really saying anything. And more importantly, the region is in that case representative for the entire country.

M&M:
The authors often use the term "blended learning approach". Would be nice for the reader to have a short summary of what this is (without the need to check references).

L94: shared with whom?

L120-122: redundant

L127: It took until the caption of fig 1 to know what NHS meant (do not think it is explained in the text)

L137-162: after reading the entire manuscript I understand better what is done but this part was really confusing. The training was conducted to improve knowledge of the vets but it was also used to develop the tool. Correct? The aim of the training was to reveal the lack of knowledge?!

Results:
L182: mentioning the 29.1% is redundant. Same as in line 193 and 195.

L198-199: I believe based on Fig 1 it is the other way around?

Fig 1: what is the value of this? This is only based on their own personal opinion? Or did you give them statements to check their actual knowledge on AMR?

Discussion:

L305: can farmers administer AMU to the animals themselves? Is not the case everywhere. Need to mention this. 

L312-313: what is the difference between the private and NHS vets? Do not think this is separated in most contries. Why is communication/collaboration so important? What does one of them have/know what the other is lacking?

Supplementary:

I do not really understand how you advise to use the tool itself. The idea is to check if all things are present/being done?
How often does a farmer/vet need to repeat this? Each time there is a problem?
And what if some things are present (eg a filter zone for changing clothes) but it is not used? How will you know?
How to practically use this is still not clear to me.

There are some sentences a bit more difficult to read. A quick proof-read would be able to solve these.

Round 2

Reviewer 1 Report

Dear authors,

thank you for addressing all my comments. I am happy now with the manuscript.